# Histone Acetylation in Central and Peripheral Nervous System Injuries and Regeneration: Epigenetic Dynamics and Therapeutic Perspectives

**DOI:** 10.3390/ijms26136277

**Published:** 2025-06-29

**Authors:** Georgina Palomés-Borrajo, Xavier Navarro, Clara Penas

**Affiliations:** 1Institute of Neurosciences, Department of Cell Biology, Physiology and Immunology, Universitat Autònoma de Barcelona, 08193 Bellaterra, Spain; georgina.palomes@uab.cat (G.P.-B.); clara.penas@uab.cat (C.P.); 2Centro de Investigación Biomédica en Red sobre Enfermedades Neurodegenerativas (CIBERNED), 28029 Madrid, Spain; 3Unitat de Neurociència Traslacional, Institut d’Investigació i Innovació Parc Taulí, 08208 Sabadell, Spain

**Keywords:** epigenetics, regeneration, histone acetylation

## Abstract

Traumatic injuries to the peripheral (PNS) and central nervous systems (CNS) trigger distinct regenerative responses, with the PNS displaying limited regenerative capacity and the CNS remaining largely refractory. Recent research highlights the role of epigenetic modifications, particularly histone acetylation, in modulating the gene expression programs that drive axonal regeneration. This review synthesizes current findings on post-translational histone modifications, focusing on histone acetyltransferases (HATs), histone deacetylases (HDACs), and epigenetic readers, in addition to their impact on neuronal and non-neuronal cells following injury. While HATs like p300/CBP and PCAF promote the expression of regeneration-associated genes, HDAC inhibition has been shown to facilitate neurite outgrowth, neuroprotection, and functional recovery in both PNS and CNS models. However, HDAC3, HDAC5, and HDAC6 demonstrate context- and cell-type-specific roles in both promoting and limiting regenerative processes. The review also highlights cell-specific findings that have been scarcely covered in the previous literature. Thus, the immunomodulatory roles of epigenetic regulators in microglia and macrophages, their involvement in remyelination via Schwann cells and oligodendrocytes, and their impact on astrocyte function are within the scope of this review. Closely considering cell-context specificity is critical, as some targets can exert opposite effects depending on the cell type involved. This represents a major challenge for current pharmacological therapies, which often lack precision. This complexity underscores the need to develop strategies that allow for cell-specific delivery or target regulators with converging beneficial effects across cell types. Such approaches may enhance regenerative outcomes after CNS or PNS injury.

## 1. Introduction

Traumatic injury to the spinal cord or the peripheral nerve disrupts the connection between the neuron somas and their target organs, leading to distal axonal degeneration. This phenomenon results in distinct functional deficits according to the nature of the injury [1]. To reverse functional impairment, research has focused on unraveling strategies that promote neuronal survival and enhance axonal regeneration, considering the differential characteristics between the peripheral nervous system (PNS) and the central nervous system (CNS).

Specifically, the PNS has the intrinsic capacity to regenerate injured axons, although the regenerative process is usually slow and incomplete [2]. Axonal regeneration in the CNS is more limited, due to an inhibitory environment formed by glial cells and an inefficient pro-regenerative program [3]. Over the past decades, researchers have focused on studying both the regenerative mechanisms active in the PNS and the limited regenerative capacity of CNS neurons. The aim was to uncover molecular pathways involved in axonal growth, including injury-signaling mechanisms, the expression of regeneration-associated genes (RAGs), the formation of a growth-supportive environment, and the presence of growth-inhibitory factors [4,5,6,7].

Those earlier findings led to the belief that combinatorial interventions on different pro-regenerative factors could lead to increased chances of axonal regeneration. Most strategies involved the modulation of transcription factors, signaling molecules, and other specific growth-related genes. While some studies supported this idea, others did not observe synergistic effects [8,9,10]. These differences may be due to the lack of accuracy in the selection of RAGs and transcription factors. For instance, a lack of consideration of time-dependent dynamics and interactions between transcription factors can significantly affect the success of regeneration in experimental settings [6,11].

Given the multitude of elements influencing axonal outgrowth, epigenetic strategies have emerged as a tool to target multiple gene sets involved in the regenerative process. Epigenetic mechanisms trigger modifications on the state of chromatin, altering gene expression transitionally or persistently. This ultimately permits adaptation to distinct stressors, like environmental challenges or injury [12]. Thus, understanding the epigenetic mechanisms involved in the response to neural injuries and in the regenerative process may lead to new successful strategies with which to enhance axonal growth. In recent years, research has started to elucidate some connections between epigenetic machinery and neuronal regeneration. The aim of this article is to summarize the current knowledge on the epigenetic modifications associated with axonal regeneration, mainly after spinal cord injury (SCI) and peripheral nerve injury (PNI), with a particular focus on post-translational histone acetylation. Although this issue has been previously reviewed [11,13,14], relevant novel findings have been reported since these publications. In addition, we also seek to highlight cell-type-specific mechanisms, which have not been considered in the earlier literature and are of importance for promoting axonal growth.

## 2. Post-Translational Histone Modifications

Nucleosomes are considered the basic unit of chromatin, consisting of DNA wrapped by histone octamers. These octamers are composed of the core histone proteins H2A, H2B, H3, and H4. Alterations in the structure of octamers can modulate the chromatin accessibility, influencing subsequent gene expression. In this regard, histones undergo post-translational modifications in their N-terminal tails, such as acetylation, methylation, phosphorylation, and ubiquitylation, among others. The post-translational marks are added by enzymes named “writers” and deleted by enzymes named “erasers”. The resulting histone code can be recognized by a group of proteins known as “readers”, which detect the post-translational marks on distinct genomic regions [12,15]. The most studied post-translational modifications are histone methylation and acetylation.

### 2.1. Histone Acetylation

Histone acetylation relaxes the interaction between the DNA strands and the histones, leading to an open chromatin configuration that facilitates transcription. The attachment of acetyl groups to histones is performed by histone acetyltransferases (HATs), whereas histone deacetylases (HDACs) remove acetylated histone marks. The newly formed acetylations or deacetylations constitute a histone code that can be recognized by reader proteins with specialized domains. Examples of these are bromodomains and chromodomains [16]. Well-known histone acetylation marks are shown in the Table 1 below.

### 2.2. Histone Methylation

Histone methylation can occur in lysine, arginine, and histidine residues. Lysine methylation has been the most studied, specifically occurring at three distinct locations: histone 3 lysine 4 (H3K4), histone 3 lysine 9 (H3K9), and histone 3 lysine 27 (H3K27) [21].

H3K4 can be mono-, di- or trimethylated [12]. Each degree of methylation has a different outcome on gene expression. Hence, monomethylated H3K4 is found in enhancers and promoters at the 3′ of active genes, facilitating gene transcription. H3K4me2 is usually found in gene bodies and enhancers that are linked with active genetic expression, whereas H3K4me3 is found in promoters of both working and dormant genes [21]. It is important to acknowledge that the same modifications can have opposing roles, such as repression and activation. This occurs with H3K4me2 and H3K4me3, that despite them usually being linked to gene activation, they can also lead to gene repression.

Contrarily to H3K4, the di- and trimethylation of H3K9 promote transcriptional repression, whereas the trimethylation of H3K27 leads to genetic repression [12,21] (Table 2).

## 3. Histone Acetylation and Its Influence on Axonal Regeneration

In this section, the roles of epigenetic writers (histone acetyltransferases), erasers (histone deacetylases), and readers will be explored to uncover their effects on axonal outgrowth, focusing on neuronal cell subtypes. It is noteworthy that epigenetic targets do not exclusively modulate histone acetylation and may also affect non-histone proteins. This is of particular interest in the case of microtubule dynamics, where its acetylation status can influence the consolidation of the growth cone and subsequent regeneration. This last concept will be further explored in Section 3.2.2, where we disclose that differences in the acetylation status of tubulin have opposite effects on the regenerative capacity of the CNS and PNS.

### 3.1. Histone Acetyltransferases (HATs)

HATs are epigenetic writers that acetylate lysine residues in histone tails and transcription factors [25]. The activity of HATs triggers a more relaxed chromatin structure, which facilitates the accessibility of the DNA favoring subsequent gene expression. Several HATs have been found to participate in axonal outgrowth, promoting recovery after traumatic nervous system injuries. Additionally, some hyperacetylated histones have been associated with axonal elongation (Table 3).

#### 3.1.1. p300/CBP

CREB binding protein (CBP) and p300 are distinct proteins that function as transcriptional coactivators with histone acetyltransferase (HAT) activity [32]. Gaub et al. observed that the overexpression of CBP/p300, p300/CBP-associated factor (PCAF), and acetylated p53 led to increased neurite growth in cultured rat cerebellum granular neurons (CGNs), both on permissive and restrictive substrates [25,26]. Mechanistically, CBP/p300 and PCAF led to increased acetylation of H3K9-14 and p53. Importantly, acetylated p53 occupancy was located on the promoters of multiple RAGs (such as *Gap43*, *Coronin1b*, *α-tubulin*, and *Sgc10*), thus increasing the expression of pro-regenerative genes. In addition, hyperacetylated H3K9-14 led to increased CBP/p300 and PCAF transcription, indirectly maintaining a regenerative loop in vitro (Figure 1) [25]. Beyond the in vitro findings, the role of p300 has been further studied in vivo. In this regard, p300 overexpression promoted axonal regeneration but did not support the survival of retinal ganglion cells after an optic nerve crush injury in rats. Specifically, p300 acetylated both CBP and p53, in addition to occupying and acetylating the promoters of the RAGs *Gap43, Coronin1b*, and *Sprr1a* [26]. Apart from the accessibility to certain RAGs, p300 has also been found to bind the promoters of the synaptic components *Psd95*, *Shank2*, and *Shank3* in cultured hippocampal neurons, suggesting that its activation has an important role in mediating synaptic plasticity [27].

Provided that CBP/p300 enhanced the expression of RAGs favoring plasticity, several authors have explored their effects as potential targets to treat SCI and PNI. For instance, resveratrol was used as a treatment to improve functional outcomes after sciatic nerve crush in rats. Resveratrol treatment improved motor performance and nerve regeneration, probably through an increase of the expression of VEGFs (VEGFb, VEGFR1, and VEGFR2). Additionally, VEGFs transcription was found to be p300-dependent, as inactivation of p300 reversed the expression of VEGFs and impaired motor recovery in animals that underwent crush injury [28]. Considering SCI, pharmacological activation of p300/CBP, using the small-molecule TTK21, led to improved functional recovery after dorsal hemisection in mice and after spinal cord contusion in rats. TTK21 treatment allowed recovery of impulse conduction across the lesion site in mice and increased the number of regenerative axons. TTK21 treatment also improved locomotion performance, which was associated with the sprouting of descending pathways below contusion SCI in rats [29]. These positive results correlated with neurite outgrowth in TTK21-treated DRG cultures [30]. TTK21 was also used as a delayed treatment in a chronic study over 22 weeks after a thoracic spinal cord transection. Treatment with the p300/CBP activator increased the expression of several RAGs in DRG neurons and yielded histological evidence of axonal regeneration and neural plasticity. Nevertheless, no functional improvement was observed in the sensorimotor tests conducted in vivo [30]. To note, in both studies TTK21 administration did not affect glial reactivity. This is of utmost importance as, in chronic experimental settings, the integrity of the glial scar may negatively affect functional outcome [29,30]. Additionally, CBP/p300 activation in the spinal cord only activated RAGs in sensory neurons, while the corticospinal neurons did not show pro-regenerative gene expression [30]. To justify these results, the authors suggested that these neurons may revert to an embryonic state after CBP/p300 activity, while sensory neurons may respond through the activation of a pro-regenerative program. These observations indicate that p300/CBP activity may lead to different consequences in regeneration of different neuronal types.

#### 3.1.2. PCAF

PCAF stands for p300/CREB-binding protein-associated factor, and is an acetyltransferase that belongs to the PCAF/Gcn5 family [33]. It was first associated with regeneration by Wong et al., who uncovered that PCAF is a downstream target of the NGF receptor [34]. Specifically, they observed that, under NGF stimulation, PC12 cells displayed neurite outgrowth, as well as nuclear translocation of the acetyltransferases PCAF and hGCN5, which were responsible for p53 acetylation [34]. Although this study established a relationship between NGF receptor activity and PCAF, it failed to demonstrate the exact influence of PCAF on neurite growth. This relationship was later evidenced by Puttagunta et al., who reported that PCAF overexpression triggers neurite growth in dissociated DRG cultures in both permissive and repressive environments [31]. Additionally, they reported that sciatic nerve axotomy led to increased H3K9 acetylation by PCAF at the promoters of the regeneration-associated genes *Gap43*, *Galanin*, and *Bdnf*. Subsequently, PCAF was overexpressed after a dorsal column lesion in the spinal cord, resulting in a significant increase in the number of regenerating fibers across the lesion site [31]. However, while PCAF was found to raise axonal outgrowth, there is no information regarding if it influences functional recovery. Further studies could answer this question by assessing nervous system conduction using electrophysiological, locomotion, and algesimetry tests to evaluate sensorimotor function in animals subjected to SCI or PNI.

### 3.2. Histone Deacetylases (HDACs)

Histone deacetylases (HDACs) are erasers that eliminate acetyl groups from histones, leading to a compacted chromatin structure. This has been classically associated with gene repression, since a closed chromatin conformation limits DNA accessibility to the transcriptional machinery. Histone deacetylases can be divided into class I (HDAC1, 2, 3, and 8), class II (HDAC4, 5, 6, 7, 9, and 10), class III (Sirtulins), and class IV (HDAC11) [35].

Since histone acetylation has been linked to plasticity and neurite growth, several authors have used HDAC inhibitors to guarantee the presence of acetyl groups bound to chromatin. This would ultimately maintain an open chromatin structure, which may allow the expression of pro-regenerative programs. Hereof, Gaub et al. observed that cortical and cerebellar neurons of mice at embryonic and postnatal stages display increased H3K9-14 acetylation when compared to adults, indicating that may be a mark for plasticity. Furthermore, they treated primary neurons with the HDAC I/II inhibitor trichostatin A (TSA), obtaining increased neurite number and length [25]. In another study, treatment with the HDACI/II inhibitor Vorinostat led to increased neurite length in Neuroscreen-1 (NS-1) cells [36]. HDAC I/II inhibitors have also been proven to be effective for regeneration in vivo, as three-day Mocetinostat treatment led to earlier axon regrowth and faster motor and sensory recovery than vehicle in mice subjected to crush injury [37]. Similar positive results have been obtained in mice that underwent dorsal column lesion and received treatment with the class I inhibitor MS-275, which specifically targets HDAC1, HDAC2, and HDAC3. Treated mice displayed increased axonal regeneration, without altering the integrity of the glial scar. However, in vitro MS-275 treatment did not increase neurite outgrowth in DRG dissociated cultures [38]. Thus, these reports highlight the involvement of class I and II HDACs in regenerative events. In the following sections, the different members of class I and II histone deacetylases will be dissected in more detail to unravel their specific effects in neural injuries and regeneration.

#### 3.2.1. Class I HDACs

Class I HDACs are ubiquitous nuclear enzymes that act as transcriptional repressors by closing chromatin configuration [35,39]. Class I HDACs have been studied in dorsal column transection and contusion SCI, identifying their expression and localization at 7 days post-injury (dpi). HDAC1 was induced at the lesion core and in axonal tracts. HDAC2 and 8 were detected at the lesion borders, whereas HDAC3 was found prominently increased at the lesion center [40]. Considering the involvement of class I HDACs in SCI, in this section we cover the in vitro and in vivo effects of HDAC1, 2, and 3 on neural regeneration.

HDAC1

HDAC1 expression dynamics of high- and low-capacity regenerating neurons have been studied after spinal cord transection in lampreys. Both neuronal subgroups downregulated HDAC1 expression early after injury, until week 4. At 10 weeks, HDAC1 expression was elevated in high-capacity neurons, whereas the low-regeneration neurons displayed an opposite behavior [41]. HDAC1 downregulation may induce dedifferentiation after injury, facilitating acetylation and the subsequent expression of pro-axonal growth genes. At the end of the regenerative program, neurons upregulate HDAC1, leading to redifferentiation and the acquisition of a mature phenotype [42]. However, targeted research of HDAC1 in mammals still needs to be assessed.

HDAC2

HDAC1 and 2 have been found to be robustly expressed after sciatic nerve crush in mice. Both targets were found increased in Schwann cells after injury, but only HDAC2 expression was initiated as early as 1 dpi and was maintained throughout the whole regenerative process. HDAC2 is a key target to regulate myelin clearance and remyelination after PNI [37]; however, its specific involvement will be further disclosed in the subchapter “Findings in non-neuronal cell types”. The same study also indicated that HDAC1/2 double knockout favored axonal growth following nerve injury [37].

Consistent with these findings, HDAC2 inhibition in the CNS has also been found to enhance plasticity. Specifically, HDAC2 knockdown in cultured cortical neurons increased synapsin punctate signals, indicating increased synapse formation. These in vitro results were also replicated in vivo. Mice subjected to traumatic brain injury (TBI) displayed increased synapse number and improved performance on functional tests after treatment with CI-994 [43]. However, CI-994 is not a specific inhibitor for HDAC2, but a whole class I HDAC inhibitor. Therefore, a specific HDAC2 knockdown followed by functional testing would be needed to uncover if the observed effects can be attributed to this target. Additionally, the study showed opposite dynamics between HDAC2 and BDNF transcription after injury. Chromatin immunoprecipitation (ChIP) experiments revealed a reduction in H4K5 acetylation at the BDNF promoter I in the denervated side of the spinal cord following TBI. These effects were compensated after HDAC2 knockdown, which leads to hyperacetylation in the BDNF promoters II and IV after injury. These results suggest that HDAC2 acts as a negative regulator of BDNF by promoting the deacetylation of its promoter region.

Given that BDNF facilitates regeneration and neuroprotection after SCI, specific HDAC2 inhibition could be a promising strategy to enhance outcome after such damage [44].

HDAC3

There is certain disagreement regarding the effects of HDAC3 inhibitors on neurite growth in vitro. Kuboyama et al. reported that HDAC3 inhibition with the specific inhibitor RGFP966 had no effects on axonal growth ex vivo, while the highest dose of 10 μM was reported to lead to a slight but not significant reduction in neurite length [40]. These results have been contradicted by Hervera et al., who reported that the same inhibitor increased neurite length in both permissive and restrictive substrates [45]. Discrepancies between studies may be attributed to the duration of the treatment and to the dissociated DRG culture experimental set-up, since both studies used the same inhibitor and concentrations. Additionally, Hervera et al. overexpressed HDAC3 by means of a viral vector, observing reduced neurite length in dissociated ganglia. Contrarily, the overexpression of an inactive HDAC3 isoform increased regeneration. Thus, although the use of inhibitors has led to conflicting results, evidence seems to point out that HDAC3 is a repressor of neurite extension. Mechanistically, the regeneration-repressive capacity of HDAC3 relies on its phosphorylation status, which is indispensable for triggering its enzymatic activity. In a physiological paradigm, calcium-dependent protein phosphatases PP4/2a dephosphorylate HDAC3, inhibiting its activity and leading to an increased acetylation state that promotes the activation of multiple pro-regenerative pathways in the DRG (Figure 2A) [45].

Considering in vivo experiments, administration of the HDAC3 inhibitor RGFP996 promoted growth of sensory fibers after dorsal column section in the spinal cord, and increased H3K9ac in DRG sensory neurons [45]. These in vivo results are consistent with another study that indicated that the inhibition of HDAC3 with RGF996 leads to functional recovery after spinal cord contusion, demonstrated by behavioral and histological data. However, in this case, recovery may also be dependent on neuroprotection and a reduction of the glial scar inhibitory substrate CSPG [40]. Likewise, positive results have been reported in rats that underwent spinal cord contusion injury and were treated with RGFP966. HDAC3-inhibited rats displayed improved BBB performance, reduced edema, and rescued blood–spinal cord barrier integrity at 7 dpi. These animals also displayed increased neuroprotection due to an arrest of the apoptotic events after treatment [46]. Nevertheless, although most studies agree on the beneficial effects of HDAC3 inhibition as a therapy to treat SCI, a study that performed a thoracic spinal cord hemisection on Balb/c mice reported that HDAC3 inhibition did not enhance functional outcomes after injury [47]. The different results may be due to the severity of the lesion, since most of the studies that report beneficial effects are based on spinal cord contusion. Additionally, the enhancement of axonal growth reported by Hervera et al. was observed on sensory fibers after dorsal cord hemisection [45]. Thus, potential differences in outcomes may rely on motor neurons, which have not been fully covered in the current studies.

#### 3.2.2. Class II HDACs

Class II HDACs are characterized by shuttling between the nucleus and the cytoplasm. This process is regulated by phosphorylation, which facilitates the binding of 14-3-3 proteins that promote the export of class II HDACs from the cytoplasm [35,39]. Among them, HDAC5 and HDAC6 proteins have been specifically studied in the context of nerve and spinal cord injuries. Additionally, class II HDACs are downregulated in several structures of the rat brain after TBI, making them attractive targets with which to enhance neural plasticity [48].

HDAC5

Microtubule dynamics are crucial for the formation of the growth cone and axonal regeneration. One of the mechanisms responsible for regulating microtubule status is acetylation, which is associated with microtubule stability [49]. In this regard, injury to PNS neurons in vitro and in vivo has been found to drastically reduce tubulin acetylation close to the lesion site, creating a distal gradient. It is noticeable that the deacetylation of the microtubules occurs in PNS neurons and fails to occur in CNS neurons. In addition to the deacetylation gradient, PNI induces an HDAC5 gradient, suggesting HDAC5 as a key element regulating regeneration in the PNS [50].

Mechanistically, in basal conditions, HDAC5 is found in the nucleus repressing the expression of RAGs by maintaining a closed chromatin conformation. After axonal injury, there is a propagation of a calcium wave throughout the nerve. This event leads to PKCμ activation, which translocates HDAC5 into the cytoplasm, leading to increased H3 acetylation levels at the nucleus. Increased acetylation in the nucleus facilitates the expression of RAGs and regeneration-associated pathways. Parallelly, PKC also phosphorylates HDAC5, favoring local tubulin deacetylation and further favoring microtubule dynamics (Figure 2A) [50,51].

Unexpectedly, while loss of HDAC5 activity impedes regeneration, it has also been found that HDAC5 overexpression represses axonal regrowth [50]. Thus, although HDAC5 is a key element of the regenerative process in the PNS, it may have a limited therapeutic window. However, other strategies that favor HDAC5 translocation, such as treatment with the PKCµ activator ingenol 3-angelate (I3A), have been successful in increasing regeneration of sensory and motor neurons after a sciatic nerve crush [51].

HDAC6

HDAC6 has also been implicated in neuroprotection and neurite growth events in vitro. Oxidative stress led to a time-dependent increase of HDAC6 in cortical neuron cultures, making it an attractive target to assess its effects on neuroprotection. Additionally, HDAC6 specific inhibition, via pharmacological treatment or with siRNA, led to neuroprotection under oxidative stress, and neurite extension under growth-repressive conditions. The mechanism appears to be transcription independent, since an increase in tubulin acetylation was found in treated cells, whereas H4 acetylation remained undetected [52]. These results suggest that microtubule stability in CNS neurons is necessary for axonal growth. Contrarily, as mentioned before, HDAC5 inhibition prevents regeneration both in vitro and in vivo due to tubulin stabilization. In this concern, it has been suggested that these differences in microtubule stability may be due to the intrinsic differences between the CNS and PNS. Thus, while in the CNS microtubule stability favors axonal growth, in the PNS less stable microtubules are required to support regeneration [50,53]. In agreement, SCI leads to decreased levels of acetylated tubulin and increased HDAC6 expression early after injury. HDAC6 inhibition using Tubastatin A improved the outcome after SCI by improving motor performance on Basso Mouse Scale (BMS) tests and footprints. The mechanism behind these effects relies on the regulation of the autophagic flux and the microtubule dynamics. In this regard, HDAC6 inhibition with Tubastatin A increases the interaction of Syntaxin17 and VAMP8, which are membrane proteins involved in the fusion of autophagosomes and lysosomes. Additionally, HDAC6 inhibition increased tubulin acetylation, favoring transport of autophagosomes and lysosomes and facilitating their fusion [54]. Importantly, recent studies suggest that autophagic defects lead to dystrophic axonal endball formation, which may impair axonal growth [55].

#### 3.2.3. Class III HDACs

Class III HDACs consist of seven sirtuins that require the NAD+ cofactor for their activity [39]. In this review, we will focus on SIRT1, as it is the most extensively studied member of the Class III family.

Sirtuin 1 (SIRT1)

Histone deacetylase SIRT1 has emerged as a critical regulator of neuronal regeneration and neuroprotection. The use of artificial intelligence analysis facilitated the screening of various FDA-approved drugs, leading to the development of a novel synergistic drug combination of acamprosate and ribavirin named Neuroheal. Neuroheal promoted functional recovery in rats subjected to nerve crush injury, and conferred neuroprotection in animals that underwent root avulsion injury. Treatment with Neuroheal resulted in improved electrophysiological recordings of muscle reinnervation and increased number of reinnervated neuromuscular junctions following nerve injury. Mechanistic studies identified that one of the primary targets of Neuroheal was SIRT1. In this context, SIRT1 overexpression in rats subjected to root avulsion resulted in a reduction in both H3K9 acetylation and p53-K373 acetylation, which correlated with enhanced motor neuron survival [56].

Beyond its neuroprotective role, SIRT1 also plays a significant role in neuronal regeneration. In this regard, in vivo overexpression of SIRT1 enhanced motor regeneration following sciatic nerve injury. Additionally, SIRT1-overexpressing rats exhibited elevated levels of ATG5-ATG12 conjugate at 7 dpi, suggesting that SIRT1 promotes autophagy. In parallel, the role of one of SIRT1 cytosolic substrates, the hypoxia-inducible factor 1-alpha (HIF1α), was explored. Accumulation of HIF1α induced neuritogenic effects dependent on autophagy. Similarly, in vitro experiments using the SH-SY5Y neuroblastoma cell line revealed that SIRT1 overexpression increased neurite length, an effect that was attenuated following HIF1α silencing. Altogether, these findings suggest that a HIF1α–SIRT1 axis regulates autophagy to facilitate neuronal regeneration [57].

Another interactor of SIRT1 is the micro-RNA miR-138, which is a repressor of sensory neuron regeneration. miR-138 interaction with SIRT1 occurs predominantly in axotomized DRGs and leads to SIRT1 repression and vice versa. Mechanistically, SIRT1 upregulation preceded miR-138 downregulation after injury. Thus, SIRT1 serves as a central promoter of axon regeneration, whereas miR-138 acts as a modulator of SIRT1 expression following axotomy. Although SIRT1 overexpression alone did not enhance axonal outgrowth in DRG neurons under standard culture conditions, it promoted neurite extension under suboptimal conditions characterized by low laminin concentration [58].

SIRT1 has also been used as a target after SCI. Activation of SIRT1 using the agonist SRT1720 improved functional outcomes, as evidenced by BMS score, reduced infiltration of inflammatory cells, and the alleviation of hemorrhage after the injury [59,60].

### 3.3. Readers of Acetylated Residues

Although most research has mainly focused on HATs and HDACs, few studies have explored the involvement of epigenetic readers in regenerative events (Table 4). Epigenetic readers recognize specific histone modifications and have the capacity to recruit protein complexes that interact with these newly formed histone marks. Special attention has been directed toward the reader family of BET proteins, as well as BRG1, a key member of the chromatin remodeler family SWI/SNF.

#### 3.3.1. BET Proteins

The bromodomain and extra-terminal domain (BET) proteins are a family of epigenetic readers capable of binding to acetylated residues linked to histone and non-histone proteins [66]. This family consists of BRD2, BRD3, and BRD4, which are ubiquitously distributed, and of BRDT, which is mainly located in germinal cells [67].

BET proteins are an attractive target for SCI treatment. In particular, the expression of the protein BRD4 has been reported to increase after spinal cord contusion in mice [61]. Consequently, several studies have used BET inhibitors to unravel their effects following SCI. BET protein inhibition using a long-term treatment regimen with the pan-BET inhibitor JQ1 improved functional outcome in mice subjected to spinal cord contusion. Specifically, JQ1 treatment led to a higher score in the BMS test and reduced neuropathic pain, due to improved neuroprotection and decreased inflammation [63]. The beneficial effects of BET inhibition have also been supported by other studies in which BMS test, BBB test, balance beam test, inclined beam test, and paw footprints were improved in mice and rats subjected to spinal cord contusion and crush, respectively [61,62]. Since immunohistochemical experiments demonstrated that BRD4 expression rose in neurons after SCI, BRD4 was also studied in vitro using the neuronal cell line PC12. Results proved that BET inhibition protected neurons against oxidative stress. Additionally, these effects were also supported by in vivo research in which recovery of autophagic flux attenuated oxidative stress and apoptosis in BET-inhibited mice after SCI [61]. Thus, BET protein inhibition promotes neuroprotection and prevents secondary injury by reducing inflammation and by regulating the autophagic flux.

BET proteins have also been studied after PNI. Inhibition of BET proteins using JQ1 led to an increment of *Gap43* and of anti-inflammatory cytokine transcription. Nevertheless, the compound was ineffective in improving the outcome after nerve injury and had detrimental effects when applied in DRG explants in vitro. Specifically, increasing doses of JQ1 decreased neurite length in a concentration-dependent manner, without inducing cellular death in DRG neurons [64]. These same effects have been reported in the cell line PC12, where treatment with JQ1 at 400 nM reduced neurite extension [61]. These observations are consistent with cortical neurons, where BET protein inhibition led to attenuated transcriptional activation and decreased expression of critical synaptic proteins [68]. Considering the PNS, there were contradicting results between in vitro and in vivo experiments. Administration of JQ1 did not influence sensory or motor reinnervation after crush injury in mice. On the other hand, whereas direct addition of JQ1 to the medium of cultured DRG neurons reduced neurite growth, the addition of a conditioned medium from JQ1-treated macrophages promoted neurite outgrowth. The authors attributed the compensatory effects to the role of BET inhibition in macrophages, which secretes pro-regenerative factors that might counteract the direct negative effects of BET inhibition on neurons [64].

Most of these studies attributed the observed effects to BRD4. This may be justified by the fact that, although JQ1 is a pan-BET inhibitor, it has a stronger preference for BRD4 [69]. Nevertheless, further research is still needed to assess, by specific silencing of BRD4, if the observed effects are due only to BRD4 or to the inhibition of multiple BET family members.

#### 3.3.2. BRG1

BRG1, also named SMARCA4, is a member of the chromatin remodeling complex SWI/SNF, which facilitates access to the DNA template for various transcription factors [70]. Specifically, BRG1 has bromodomains that allow its binding to acetylated lysine residues of H3 and H4, although the interaction is weak [71]. In experimental settings, *Brg1* deletion in hippocampal neurons reduced dendritic spine density and maturation, and impaired excitatory synapse transmission both in vitro and in vivo. Additionally, BRG1 was found to be responsible for the regulation of genes that encode channels, neurotransmitter release, and synaptogenesis [65]. Altogether, these data suggest that BRG1 is a key element for neuronal plasticity, an event of utmost importance after SCI. Regarding the PNS, BRG1 has been found to regulate Schwann cell differentiation [72], findings that are covered in the next subsection.

## 4. Epigenetic Modifications in Non-Neuronal Cells and Consequences for Neural Regeneration

Glial cells play a critical role in the acute response to injury and in shaping the microenvironment for subsequent axonal regrowth. For instance, macrophages and Schwann cells are involved in the clearance of cellular debris following nerve injury. These cells remove inhibitory components that impede axonal regeneration [73]. Astrocytes acute responses rely on the formation of the glial scar that confines the neural lesion to a localized area, preventing the expansion of neurotoxicity. Microglia also contribute to debris clearance in the injured CNS. However, glial cells also have detrimental effects. Excessive activation of microglia and astrocytes can lead to chronic neuroinflammation and the release of factors that hinder neuronal survival as well as axonal regrowth [74]. Therefore, a deeper insight into the epigenetic modifiers in these cells may help to prepare strategies to enhance their beneficial effects on regeneration while mitigating their negative impact, ultimately supporting functional recovery (Figure 2B and Figure 3).

### 4.1. Myeloid Cells: Microglia and Macrophages

Histone acetylation is a key element in the orchestration of early and late inflammatory genes, and to the response of environmental factors [75,76,77]. Given that inflammation depends on acetylation and that inflammatory events are of utmost importance after PNI and SCI, the influence of acetylated histone modifiers and readers in macrophages and in microglia is also highly relevant (Figure 3A).

Histone Deacetylases and the Relevance of HDAC3 in Inflammatory Events

Although multiple histone deacetylases have been associated with the regulation of inflammatory processes, the specific contributions of individual HDAC members following SCI or PNI remain poorly understood [78]. From the histone deacetylases, HDAC3 has been extensively studied in the context of post-SCI inflammation. HDAC3 accumulates at the lesion epicenter, peaking at 7 dpi after SCI in mice. HDAC3 was colocalized with myeloid markers and occurred more prominently in pro-inflammatory cells, which exhibited lower levels of AcH3 compared to the cells with an anti-inflammatory phenotype. By inhibiting HDAC3 with RFP966 acutely after a contusion injury, there was a decrease in the number of pro-inflammatory cells without altering the influx and proliferation of immune cells in the spinal cord. Additionally, HDAC3 inhibition produced inflammatory suppression, detected by a cytokine/chemokine antibody array [40]. Similar results have been reported in rats, in which HDAC3 inhibition significantly decreased microglial activation and expression of the inflammatory cytokines TNF-α, IL-1β, and IL-6 after SCI. In addition, HDAC3 inhibition with RFP966 diminished oxidative stress by regulating HO-1 and NQO-1 expression through Nrf2 [46]. Contrarily, a study conducted on macrophages in vitro indicated that RGFP966 did not affect the expression of *Il-6* and *Il-1β* after LPS stimulation. However, this study reported that HDAC3 inhibition promoted the gene expression of M2 markers after IL-4 or IL-13 stimulation. Nevertheless, it did not affect functional recovery after SCI [47].

Concordant with the above results, transcriptome analysis showed that microglia and macrophages shared HDAC3-dependent genes associated with immunological function. These genes belong to chemotaxis, chemokine activity, p38 MAPK, and receptor activator of nuclear factor kappa-B ligand (RANKL) pathways. Additionally, single-cell RNA sequencing data have revealed four transcriptional clusters of microglia after injury: immunity MG1 (cytokine expression), reactive MG2 (phagocytosis markers), immediate-response MG3, and proliferative MG4. The four clusters in control animals consisted of MG1 (55%), MG2 (5%), MG3 (25%), and MG4 (15%). These ratios were altered after SCI and in animals that received an HDAC3 inhibitor after injury. Specifically, SCI led to an increase in MG1 and MG2, reaching 67% and 22%, respectively. Contrarily, MG3 and MG4 populations decreased after trauma, representing 8% and 2%, respectively, of the microglial cells at 5 dpi. Treatment with the HDAC3 inhibitor modified these changes and promoted slight recovery towards the homeostatic state, as the MG2 marker decreased to 17% after treatment and MG4 increased to 5% [79].

Altogether, these findings point out that HDAC3 inhibition has a repressive immunological effect. Moreover, many of these studies reported improved functional outcomes after SCI, attributing these benefits in part to HDAC3’s role in modulating inflammation and secondary injury [40,46,79].

**Figure 3 ijms-26-06277-f003:**
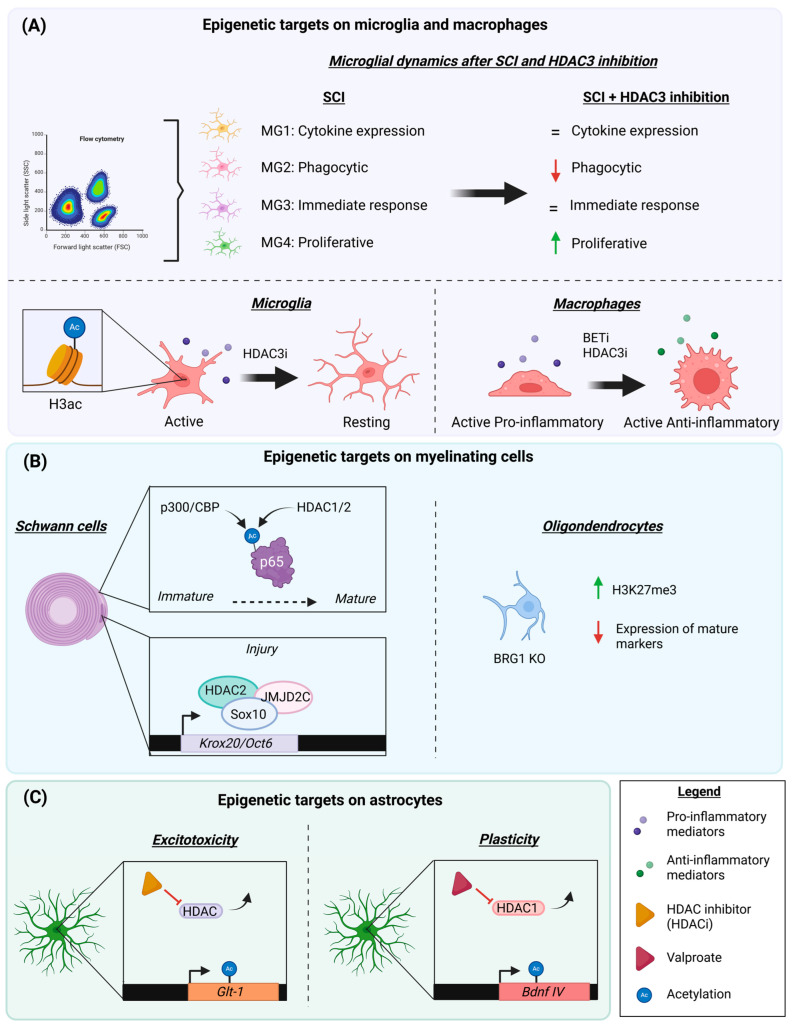
(**A**) Microglia and macrophage functions are modulated by HDAC3 and BET proteins. Microglial transcriptional clusters after SCI are altered upon HDAC3 inhibition [79]. HDAC3 inhibition reduces the expression of inflammatory mediators, suppressing the immune response [40,46]. HDAC3 and BET protein inhibition favor the expression of anti-inflammatory mediators on macrophages, acting as an inflammatory switch [63,64]. (**B**) HDAC1/2 is essential for the maturation and myelination of Schwann cells. After injury, HDAC2, Sox10, and JMJD2C are recruited to the myelination genes *Oct6* and *Krox20* to demethylate H3K9 and activate gene expression in a timely manner [37]. BRG1KO indirectly increases H3K27me3 expression, repressing the expression of maturation and myelination markers in oligodendrocytes [80]. (**C**) HDAC activity represses the expression of *Glt-1* and *Bdnf* genes. Upon HDAC inhibition, astrocytes enhance the expression of *Glt-1* and *Bdnf*, reducing excitotoxicity and creating a microenvironment that favors neural plasticity [81,82].

BET Proteins and Inflammation

The epigenetic reader family of BET proteins has largely been associated with the regulation of inflammatory events [83]. Following SCI, the inhibition of BET proteins with JQ1 led to the decreased transcription of the pro-inflammatory mediators *Il-6, Il-1β*, *Tnf-α*, and *Ccl2* at distinct time points after the insult. Contrarily, BET protein inhibition led to a rise in the transcription of the anti-inflammatory cytokine genes *Il-4, Il-10*, and *Il-13*. The same results were replicated for the protein expression of the cytokines IL-6, IL-10, and IL-13 at 4 h after SCI. Considering these effects, treatment with JQ1 led to a reduced immune response and increased neuroprotection after SCI [63]. These results are consistent with other experiments in mice and rats, showing a fall in the expression of most pro-inflammatory cytokines and a reduced number of immune cells infiltrating the SCI site [62,84]. Similar results were found in peripheral nerves, where the early inhibition of BET proteins led to the diminished infiltration of immune cells after injury, whereas delayed BET inhibition increased the expression of the anti-inflammatory mediators *Il-4* and *Il-13* after nerve crush [64]. In conclusion, BET inhibition affects the whole inflammatory response, which is critical for functional outcome following SCI and PNI. Additionally, BET inhibition negatively affected neurite growth on DRG explants, whereas the influence of conditioned media from BET-inhibited macrophages enhanced neurite growth. We partially attributed these positive effects on neurite outgrowth to the role of anti-inflammatory mediators and the activation of the STAT6 pathway [64]. Anti-inflammatory cytokines have been associated with enhanced nerve regeneration, and STAT6 activation has been shown to promote retinal axon growth in rats [85,86,87]. These findings suggest that promoting an anti-inflammatory environment may support regeneration. Further, while the initial release of pro-inflammatory cytokines after injury is necessary for debris clearance during Wallerian degeneration, prolonged inflammation impairs the nerve regenerative capacity [88]. This study highlights that the same epigenetic target can trigger variable effects depending on the type of cell involved and the need to investigate cell-specific actions of epigenetic regulators.

### 4.2. Myelinating Cells: Schwann Cells and Oligodendrocytes

Influence of Histone Deacetylases

The role of myelinating cells can also affect the outcome of regenerative events. In this regard, histone deacetylases have been found to regulate myelination.

HDAC1/2 double knockout (KO) resulted in increased axonal growth shortly after injury, but was followed by defects in myelination at later stages. In particular, transgenic animals displayed reduced Oct-6 expression, which is necessary for the conversion of Schwann cells into a repair phenotype, and diminished Krox20, which is needed for remyelination. Altogether, these results suggested that HDAC1/2 negatively regulated axon growth but promoted remyelination. Thus, acute treatment with the HDAC1/2 inhibitor Mocetinostat after sciatic nerve crush significantly improved functional outcomes compared to vehicle animals without affecting the myelin sheath. Mechanistically, HDAC2 creates a protein complex with JMJD2C, KDM3A, and Sox10 that interacts with *Oct6* and *Krox20* genes. Although the double KO mice displayed increased H3 acetylation in the nerve, there was impaired transcription of both *Oct-6* and *Krox20*. This indicated that other events were regulating gene expression. The complex also modulated H3K9 methylation levels, which appeared as responsible for alterations in the expression of Schwann cell-related genes (Figure 3B) [37].

These results have been supported by other studies, which point out that double KO animals display a reduction in mature myelin components like *Mbp, Mag*, and *Mpz*, alterations attributed to defects in Schwann cell differentiation. HDAC1/2 regulates the maturation of Schwann cells, since the double KO animals presented a decrease in positive regulators of Schwann cell differentiation and increased negative regulators [89,90]. During development, control mice exhibited a decline in acetylated p65 levels, which was more prominent during immature stages. Indeed, in differentiated Schwann cells, the association of p65 with the histone acetyltransferase p300/CBP decreased, while its interaction with HDAC1/2 increased [89]. In addition to this mechanism, the epigenetic mark H3K4me3 was found in pro-differentiation genes, whereas the repressive mark H3K9me3 was located on genes that inhibit Schwann cell maturation. Additionally, HDAC1 and HDAC2 can compensate each other’s loss while having distinct roles in Schwann cell physiology. In fact, HDAC2 is more related to myelination, whereas HDAC1 participates in cell survival [90]. HDAC3 also regulates the physiology of myelinating Schwann cells; however, different effects have been reported regarding its specific mechanisms of action. While some findings suggest that HDAC3 increases myelination, others indicate that it contributes to restoring Schwann cells to a homeostatic state, thereby preventing hypermyelination [91,92].

Considering histone deacetylases in oligodendrocytes, SIRT1 inactivation contributes to the proliferation of oligodendrocyte progenitor cells (OPCs). Additionally, transgenic animals with SIRT1 knockdown enhance remyelination in the corpus callosum after lysolecithin-induced demyelination [93].

BRG1

BRG1 loss impairs oligodendrocyte differentiation in the brain and, to a lesser degree, in the spinal cord. In BRG1-inducible KO animals, the percentage of myelinated axons was reduced by three- to four-fold in the corpus callosum, whereas the decrease was two-fold in the spinal cord. This provides evidence that BRG1 plays a pivotal role in CNS myelination. In fact, a spinal cord demyelinating lesion caused by l-α-Lysophosphatidylcholine injection led to diminished MBP immunofluorescence and a lower number of myelinated fibers in BRG1 KO compared to controls. Gene Set Enrichment Analysis (GSEA) revealed that BRG1 KO downregulated genes associated with oligodendrocyte differentiation, such as, the expression of mature markers like *Mbp*. Of note, BRG1-KO led to the enrichment of gene signatures of H3K27me3 and PCR2. Moreover, BRG1 interacts with PRC2 (polycomb-repressive-complex) to enhance H3K27me3-mediated repression at gene loci associated with the inhibition of oligodendrocyte differentiation (Figure 3B) [80]. BRG1 is also required for Schwann cell differentiation and myelin formation, as its loss prevented myelination in the PNS [72].

### 4.3. Astrocytes

Astrocytes have a critical protective role following SCI, reducing lesion spreading and protecting cells from injury-induced excitotoxicity as well as hypoxia [94,95]. While these acute mechanisms limit tissue damage, astrocytes also secrete inhibitory molecules that impede axonal regeneration. In addition, their dysregulated responses may trigger microglia activation and disrupt the homeostatic equilibrium of the neural environment [74]. This subchapter summarizes current evidence on the function of some epigenetic targets on astrocytes, focusing on the effects on excitotoxicity, axonal growth, and neuroinflammation (Figure 3C).

HDACs in Excitotoxicity and Neuroprotection

HDAC inhibition with valproic acid has been reported to reduce astrocytic death in vitro after hypoxia induction and to attenuate astrocyte reactivity in vivo after stroke [96]. In models of SCI, valproate administration partially prevented the loss of spinal cord tissue, limiting myelin and axonal loss, and increased the number of surviving oligodendrocytes, resulting in better recovery of locomotor activity in rats [97]. Similarly, valproic acid also improved neurological test performance after TBI in swine, likely by decreasing astrocytic and microglial reactivity [98].

These neuroprotective effects of HDAC inhibitors may be explained by their capacity to mitigate excitotoxicity. In fact, pre-treatment with a class I HDAC inhibitor, MS-275, in mouse optic nerves (MONs) subjected to oxygen and glucose deprivation (OGD) led to significantly increased GLT-1 levels and decreased GFAP expression without altering astrocyte number. This treatment also delayed and reduced glutamate release in MONs of animals that received MS-275 treatment [99]. OGD conditions were used to mimic stroke, but focal hypoxia also occurs after SCI, so this model may also provide information that may be potentially applied to SCI. Supporting this, a study based on epilepsy unraveled by ChIP showed that HDAC class I/II inhibition by TSA increased GLT1/EAA2 expression through H4 acetylation [81]. Thus, HDACs may be involved in the repression of GLT-1 transcription, favoring increased excitotoxicity due to poor glutamate clearance. However, further research is needed to fully validate these results in the context of SCI, and to define specific targets implicated in excitotoxicity. Most of the studies to date have employed broad-spectrum HDAC inhibitors, making it difficult to determine the exact targets involved in excitotoxicity regulation.

Growth and Plasticity: the Importance of BET Proteins and HDACs

The specific inhibition of HDAC3 using RFP966 has been shown to reduce glial scar formation after SCI, along with a decrease in the accumulation of the growth-inhibitory molecule Condroitin Sulfate Proteoglycan (CSPG) [40]. Additionally, the HDAC inhibitors valproic acid, sodium butyrate, and Trichostatin A have been found to induce the expression of neurotrophic factors such as GDNF and BDNF in primary cultured astrocytes, an effect associated with increased H3 acetylation [100]. Elevated BDNF expression has also been detected after valproate treatment in swines that underwent TBI [98].

A growth-permissive environment also occurs via BET protein inhibition.Inhibition of BET proteins enhanced synapse formation when human IPS-derived neurons were co-cultured with IPS-derived astrocytes [68]. In vivo, treatment with the BET inhibitor JQ1 raised the transcription of the synaptic markers *Psd95* and *Synaptophysin* in animals with Aβ_1-42_ induced Alzheimer’s disease. This increase was reverted in rats that were treated with the specific astrocyte inhibitor fluorocitrate (FC) [101]. However, it is worth noting that the FC concentrations used were under 250 μM, which has been reported to affect neuronal function [102]. Altogether, these results point out that BET protein inhibition in astrocytes may help to create a favorable environment for plasticity after CNS injury.

Additionally, HDACs participate in the regulation of BDNF transcription. For instance, valproate has been shown to increase in HDAC1 protein levels, which in turn reduces H3K9 acetylation at the *Bdnf* promoter IV, and subsequent protein expression [82].

HATs and HDACs in Inflammation

Considering inflammation, the acetyltransferase p300 has been found to play a key role in the response of human primary astrocytes. Specifically, the decreased phosphorylation of p300 led to the diminished acetylation of p65 at its K310 residue compromising transcriptional activity of inflammatory mediators [103]. Given that promoter acetylation is also crucial for the expression of inflammatory genes, further research should determine whether p300 directly influences promotor acetylation at this loci [104].

Another mediator that has been recently associated with the inflammatory function of astrocytes is SIRT1. The anti-inflammatory compounds TMPZ and AGS-IV led to the increased expression of SIRT1 in cultured rat astrocytes, promoting a shift from the pro-inflammatory A1 to the neuroprotective A2 phenotype. Conversely, pharmacological inhibition of SIRT1 using EX-527 led to the prevalence of the A1 phenotype and increased pro-inflammatory gene expression. These findings suggest that SIRT1 plays an important role in regulating astrocyte inflammation, potentially through the OIP5-AS1-miR-34a-SIRT1-NF-κB signaling axis [105].

## 5. Conclusions

In this review we have addressed the current findings of post-translational histone acetylation and its influence on regeneration following traumatic nerve and spinal cord injuries. Over the past decades, significant advances have been made uncovering the link between epigenetic modifications and gene expression networks that impact regenerative outcomes. Nonetheless, several key questions remain to be addressed.

Successful regeneration after PNI or SCI is a multifaceted process influenced by several factors, including the differences between neuronal subtypes. In fact, it has been demonstrated that sensory and motor neurons exhibit different responses and rates of axonal growth after nerve injury, which can significantly impact the efficacy of functional regeneration [106,107]. In this regard, a transcriptomic study has further highlighted the differences in gene expression among various neural subpopulations following nerve injury [108]. These studies indicate the existence of neuronal-specific programs that, if modulated effectively, could promote more balanced regeneration. Some of these differences may involve epigenetic machinery. For instance, CBP/p300-mediated activation of a pro-regenerative program affected sensory neurons, whereas neurons of the corticospinal tract remained unaffected [30]. Thus, understanding intrinsic differences between neuronal types and identifying common epigenetic modifications could be essential for optimizing functional outcomes after neural injuries.

In addition, it is crucial to uncover the effects of histone acetylation and other epigenetic modifications across different cell types. Such understanding may enable bettertargeted and more efficient therapeutic strategies, reducing potential side effects, as broad-spectrum epigenetic interventions have shown undesirable effects in clinical trials [109,110]. Caution must also be taken when interpreting findings from in vitro experiments. In this regard, the use of co-culture systems should be extended, as monocultures may yield biased or incomplete information. A clear example involves BET proteins, which displayed detrimental effects on neurons in monoculture but enhanced neurite outgrowth and synapse formation when neurons were co-cultured with macrophages and astrocytes [64,68].

Despite the significant advancements in understanding epigenetic mechanisms, the development of technologies capable of inducing adequate epigenetic modifications remains in its infancy. This limitation stems from several factors, including the importance of precise timing, the complexity of disentangling co-occurring epigenetic marks, and challenges related to targeted delivery, factors that compromise their safety and prevent clinical translation [111,112]. In summary, recognizing the differences in neural and cellular subtypes and overcoming the associated challenges in their assessment will be essential for advancing in therapeutic strategies aimed at enhancing recovery following neural injuries.

## Figures and Tables

**Figure 1 ijms-26-06277-f001:**
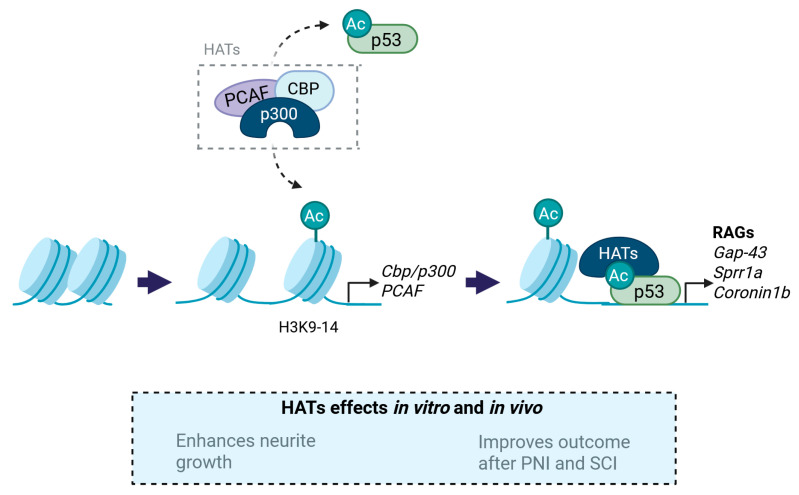
Histone acetyltransferases (HATs) promote the acetylation of both histones and p53. Hyperacetylation of H3K9-14 enhances the transcriptional activity of CBP/p300 and PCAF, maintaining a regenerative feedback loop. In addition, HATs and acetylated p53 can bind to the promoters of several RAGs, facilitating their expression. RAGs, regeneration-associated genes.

**Figure 2 ijms-26-06277-f002:**
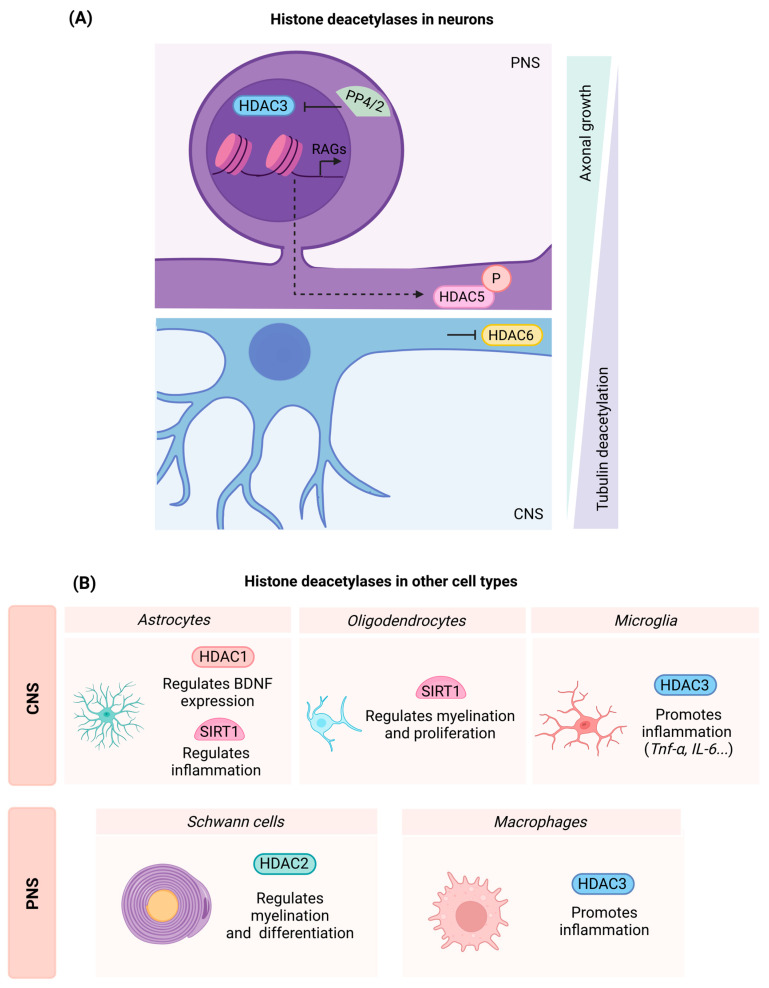
(**A**) Histone deacetylases play distinct roles in the peripheral nervous system (PNS) and central nervous system (CNS). After injury in the PNS, a calcium wave initiates the activation of PP4/2 and the translocation of HDAC5 to the cytoplasm. PP4/2 activation prevents HDAC3 activity, thereby facilitating the expression of regeneration-associated genes (RAGs). The cytoplasmatic translocation of HDAC5 also contributes to chromatin relaxation within the nucleus and promotes tubulin deacetylation in the cytoplasm, resulting in a dynamic cytoskeleton essential for PNS regeneration. In contrast, regenerative processes in the CNS rely on cytoskeletal stability, which is supported by the inactivation of HDAC6. (**B**) Histone deacetylases effects on glial and myeloid cells have a crucial impact on homeostasis and after damage in both the PNS and CNS. HDAC3 and SIRT1 are associated with inflammation, whereas SIRT1 and HDAC2 participate in myelination. Finally, HDAC1 promotes BDNF expression in astrocytes, being responsible for regulating a positive microenvironment for plasticity and axonal growth.

**Table 1 ijms-26-06277-t001:** Common histone acetylation marks and their location.

Histone Modification	Effects on Gene Expression	Location	Source
H3K9ac	Activation	Promoters, enhancers	[17]
H3K27ac	Activation	Enhancers, promoters	[17,18]
H4K8ac	Activation	Promoters	[19]
H4K16ac	Activation	Promoters, gene bodies	[19,20]

**Table 2 ijms-26-06277-t002:** Common histone methylation marks and their location.

Histone Modification	Effects on Gene Expression	Location	Source
H3K4me1	Activation	Enhancers	[18,21]
H3K4me2	Activation/repression	Gene bodies	[21]
H3K4me3	Activation or poised expression	Promoters	[21]
H3K9me2	Repression	Transposable elements, satellite repeats, and gene bodies	[22]
H3K9me3	Repression or poised expression	Enhancers, telomeres, transposable elements, gene bodies, and satellite repeats	[18,22,23]
H3K27me3	Repression	Enhancers, promoters	[18]
H3K36me3	Activation	Gene bodies	[24]

**Table 3 ijms-26-06277-t003:** Histone acetyltransferases (HATs) involved in axonal regeneration. The table summarizes histone targets, effects on gene expression, and effects on regeneration, based on findings from in vivo and in vitro studies. CGNs, cerebellum granular neurons; DRG, dorsal root ganglia; PNI, peripheral nerve injury; and SCI, spinal cord injury.

Enzyme	Hyperacetylated Histone	Altered Genes	Outcome In Vitro	Outcome In Vivo	Source
p300/ CBP	H3K9-14ac H4K8ac H3K27ac	*Gap43*, *Coronin1b*, *Sgc10*, *α-tubulin*, *Sprr1a*, *Vegfr1*, *Vegfr2*, *Vegfb*, *Psd95*, *Shank2*, and *Shank3*	Neurite outgrowth in CGNs and DRG primary cultures	Neurite outgrowth after optic nerve crush Motor function repair after PNI Functional motor and sensory recovery after acute SCI	[25,26,27,28,29,30]
PCAF	H3K9ac	*Gap43*, *Bdnf* and *Galanin*	PCAF-AVV leads to increased neurite growth in dissociated DRG cultures	Overexpression leads to histological evidence of axonal regeneration after SCI	[31]

**Table 4 ijms-26-06277-t004:** Readers associated with neural regeneration. The table summarizes direct or indirect effects on gene expression, as well as their effects on regeneration, based on findings from in vivo and in vitro studies.

Enzyme	Altered Genes	Altered Proteins	Outcome In Vitro	Outcome In Vivo	Source
BET proteins	*Gap43*, *Il-4*, and *Il-13*	IL-6, IL-10, IL-13, SOD1, CytC, HO-1, Cleaved caspase 3, Beclin, and LC3II/I	BET-inhibition improved resistance to ROS Inhibition of BET proteins prevented neurite growth	Inhibition of BET proteins improves functional outcome after SCI, but not after nerve injury	[61,62,63,64]
BRG1	*VGluT2, Camk4, Gap43*, and *Cacng3*, among others	-	Deletion reduces spine density and neuronal function	Not assessed	[65]

## Data Availability

The data supporting this review article have been published.

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
