# Peer review of "Histone Acetylation in Central and Peripheral Nervous System Injuries and Regeneration: Epigenetic Dynamics and Therapeutic Perspectives"

_ijms, 2025, doi:10.3390/ijms26136277_

Round 1
Reviewer 1 Report
Comments and Suggestions for Authors
The authors have reviewed the role of histone acetylation, focussing on acetylation, and the proteins regulating these modifications, during both peripheral and central nervous system regeneration after injury. as a topic of interest. Since histone acetylation is dynamic, and various therapeutic strategies are being developed to modulate these dynamics, synthesising the existing knowledge on the effects of histone acetylation on the regenerative capacity of the nervous system is certainly of use to the field. A number of reviews have already addressed the role of epigenetics in axon regeneration, however the current review importantly builds on these by considering cell type specific histone modifications in non-neuronal cells (macrophages, glial cell types) and non-neuronal pathways (inflammation) relevant to nervous system injury. In particular for considering therapeutic potential the findings in different cell types must be considered together. Since this is the novel aspect of the review, it would be more impactful if more weight were given to the cell type specificity sections of the review.
Much of the review content on HDACs in neurite outgrowth is based on research from before 2020 and it does not appear that there is much more recent literature on this specific aspect. More recent publications are included in the sections which add to previous reviews, on non-neuronal cell types and immunomodulatory effects of epigenetics in regeneration. I think this review would be stronger if these areas were highlighted more throughout.
Lines 293-297, on tubulin acetylation in the growth cone might be more appropriate in the introduction to section 3 as I don’t think it is specific to HDAC5, and also nicely sets out some of the difference between PNS and CNS. This difference is again a more novel aspect of the review and moving it out of a specific subsection could highlight this for readers.
Line 312 “other strategies that favour HDAC5 translocation have been successful”; it would be good to include more detail about these strategies.
Authors appear to have focussed on neurite outgrowth as a measure of nervous system regeneration, and rightly point out that this doesn’t indicate functional recovery. There is a (more recent) study which has investigated functional recovery that could add to the review as they do offer functional understanding on the effect of HDAC inhibitors in regeneration after nervous system injury; eg https://www.nature.com/articles/s41419-020-02897-w
Another recent references that might add to the review;
https://www.nature.com/articles/s41380-021-01369-7?fromPaywallRec=false
Figure 2 is clear and the concept is helpful for the reader. However, it could include far more detail, the “other cell types” should be properly labelled, and it should be referenced throughout the text.
The conclusions of the review set out clearly and concisely the directions that the field of study could take to better understand epigenetic dynamics in more complex model systems, to incorporate the effects of non-neuronal cell types and improve the likelihood of being able to promote nervous system regeneration for therapeutic benefit. In summary, this review presents novel synthesis of the literature on the role of histone modifications in non neuronal cell types during nervous system regeneration and could be impactful for the field, especially with some more focus given to the aspects of the review that are novel.
Author Response
The authors have reviewed the role of histone acetylation, focussing on acetylation, and the proteins regulating these modifications, during both peripheral and central nervous system regeneration after injury. as a topic of interest. Since histone acetylation is dynamic, and various therapeutic strategies are being developed to modulate these dynamics, synthesising the existing knowledge on the effects of histone acetylation on the regenerative capacity of the nervous system is certainly of use to the field. A number of reviews have already addressed the role of epigenetics in axon regeneration, however the current review importantly builds on these by considering cell type specific histone modifications in non-neuronal cells (macrophages, glial cell types) and non-neuronal pathways (inflammation) relevant to nervous system injury. In particular for considering therapeutic potential the findings in different cell types must be considered together.
R: We appreciate the positive consideration of this review paper by the reviewer.
Since this is the novel aspect of the review, it would be more impactful if more weight were given to the cell type specificity sections of the review.
Much of the review content on HDACs in neurite outgrowth is based on research from before 2020 and it does not appear that there is much more recent literature on this specific aspect. More recent publications are included in the sections which add to previous reviews, on non-neuronal cell types and immunomodulatory effects of epigenetics in regeneration. I think this review would be stronger if these areas were highlighted more throughout.
R: In response to the reviewer comment, we have increased the importance of the cell specificity section. The section 4 of the paper is now devoted to the role of non-neuronal cell types. We made a brief introduction to that subsection that underlines their importance and points out that this information has been limitedly covered by previous reviews. Particularly we have expanded the part dedicated to epigenetic mechanisms in astrocytes (subsection 4.3). These cells are crucial in the response after SCI and the information in the previous version of the paper was limited. In addition, within the abstract we have mentioned that cell-specific studies are reviewed.
Lines 293-297, on tubulin acetylation in the growth cone might be more appropriate in the introduction to section 3 as I don’t think it is specific to HDAC5, and also nicely sets out some of the difference between PNS and CNS. This difference is again a more novel aspect of the review and moving it out of a specific subsection could highlight this for readers.
R: We made an introduction to section 3, where we commented that it will be centered on the role of writers, readers and erasers in neuronal cells. Then, we included that epigenetic targets can also affect non-histone proteins and introduced that tubulin dynamics are also affected by this process highlighting the different outcome between PNS and CNS.
We expect that these changes facilitate reading and make stand out the strengths of this review.
Line 312 “other strategies that favour HDAC5 translocation have been successful”; it would be good to include more detail about these strategies.
- Thank you for this observation. We have included a brief indication of the compound used.
Authors appear to have focussed on neurite outgrowth as a measure of nervous system regeneration and rightly point out that this doesn’t indicate functional recovery. There is a (more recent) study which has investigated functional recovery that could add to the review as they do offer functional understanding on the effect of HDAC inhibitors in regeneration after nervous system injury; eg https://www.nature.com/articles/s41419-020-02897-w
Another recent references that might add to the review;
https://www.nature.com/articles/s41380-021-01369-7?fromPaywallRec=false
R: We have included your suggested references, as these studies strengthen some findings related to the function of neuronal HDAC2 and bring insights into their effect on functional recovery. However, we included limited research in areas such as TBI and stroke since the primary focus of the review is SCI and PNI.
Figure 2 is clear and the concept is helpful for the reader. However, it could include far more detail, the “other cell types” should be properly labelled, and it should be referenced throughout the text.
R: We have included more details on other cell types, beyond neurons, as well as proper labeling to improve clarity. We also referenced the figure in the text.

Reviewer 2 Report
Comments and Suggestions for Authors
The present manuscript is a rigorous and exhaustive examination of histone acetylation's role in nervous system regeneration, with specific differentiation between the central and peripheral nervous systems. The review uses current findings and skillfully bridges molecular, cellular, and functional elements. Some areas require further explanation, better critical synthesis, and redundancy emphasis and overall unity. The figures presented are helpful but can be maximized for informational purposes. There are also minor grammatical errors, and reference style needs to be scrutinized.
1. The title should indicate that it is a narrative review and stress the dual focus on CNS and PNS, e.g., "Histone Acetylation in CNS and PNS Regeneration: Epigenetic Dynamics and Therapeutic Implications."
2. The abstract is too descriptive. It states what is covered but does not cover essential insights. Suggest adding one or two sentences of main conclusions or controversies.
3. Lines 32–44 recapitulate familiar PNS/CNS regenerative disparities. Think about condensing or connecting them more analytically.
4. Throughout the manuscript, experiments are described but rarely critically evaluated. For example, conflicting results with HDAC3 inhibitors are presented (p. 8–9), yet the authors do not suggest which experimental conditions would be responsible for differences.
5. The HDAC sections are extremely detailed, while sections such as SIRT1 (3.2.3) and BRG1 (3.3.2) seem relatively short. Try to balance the depth.
6. The schematic diagrams are visually clean but lack panel labels or explanatory legends that could help readers decipher mechanisms between cell types.
7. The dual nature of BET inhibition (inflammation good, neurite growth bad) is presented, but no attempt is made to resolve or account for this paradox.
8. The conclusion mentions briefly delivery and timing concerns, but the review would be strengthened by discussing in greater detail the translational failures of HDAC inhibitors and BET inhibitors in the clinic.
9. Table 3 uses abbreviations like CGNs and PNI without first defining them, define all acronyms upon first use in text and tables.
10. Line 105: "as well as in distinct transcription factors" could be rephrased for clarity. Suggest proofreading in general.
11. The review mainly cites animal studies. State briefly whether there are any relevant human postmortem or iPSC-derived neuron studies.
12. The section on astrocytes (3.4.3) is not well developed. Given the significance of glial scarring to CNS repair, the role of histone modifications in reactive astrocytes needs to be discussed more completely.
13. The final part may contain a schematic diagram or decision tree of possible therapeutic approaches that are directed by epigenetic targets and by neuronal subtypes.
Comments on the Quality of English LanguageCertain sentences are too lengthy or reiterate details provided previously. Condensing them would enhance readability.
Author Response
Comments and Suggestions for Authors
The present manuscript is a rigorous and exhaustive examination of histone acetylation's role in nervous system regeneration, with specific differentiation between the central and peripheral nervous systems. The review uses current findings and skillfully bridges molecular, cellular, and functional elements. Some areas require further explanation, better critical synthesis, and redundancy emphasis and overall unity. The figures presented are helpful but can be maximized for informational purposes. There are also minor grammatical errors, and reference style needs to be scrutinized.
R: We appreciate the positive consideration of this review paper by the reviewer.
- The title should indicate that it is a narrative review and stress the dual focus on CNS and PNS, e.g., "Histone Acetylation in CNS and PNS Regeneration: Epigenetic Dynamics and Therapeutic Implications."
R: We have reformulated the title in line with the suggestion.
The new proposed title is “Histone acetylation in central and peripheral nervous system injuries and regeneration: epigenetic dynamics and therapeutic perspectives”
- The abstract is too descriptive. It states what is covered but does not cover essential insights. Suggest adding one or two sentences of main conclusions or controversies.
R: Thank you for your comment. We have modified the Abstract, particularly in the second half by adding some conclusions about the importance of considering cell-specificity and the limitations of unspecific pharmacological therapies.
- Lines 32–44 recapitulate familiar PNS/CNS regenerative disparities. Think about condensing or connecting them more analytically.
R: We appreciate the reviewer’s comment and agree that this section could benefit from greater analytical focus. In response, we have revised lines 32–44 to present a more concise and integrated comparison between PNS and CNS regenerative capacities.
- Throughout the manuscript, experiments are described but rarely critically evaluated. For example, conflicting results with HDAC3 inhibitors are presented (p. 8–9), yet the authors do not suggest which experimental conditions would be responsible for differences.
R: We appreciate the reviewer´s observation and agree that critical evaluation of discrepancies across studies is essential. Throughout the manuscript, we have aimed to highlight and contextualize conflicting findings by considering methodological or model differences between studies.
Regarding the specific case of HDAC inhibitors, we addressed the following in vitro and in vivo discrepancies in the text (p.9) by including:
- In vitro: Discrepancies between studies may be attributed to the duration of the treatment and to the dissociated DRG culture experimental set-up, since both studies used the same inhibitor and concentrations, as indicated in the text
- In vivo: Although most studies coincide on the beneficial effects of HDAC3 inhibition as a therapy to treat SCI, a study that performed a thoracic spinal cord hemisection on Balb/c mice reported that HDAC3 inhibition did not produce enhanced functional improvement. The different results may be due to the severity of the lesion, since most of the studies that report beneficial effects are based on spinal cord contusion models. Additionally, the beneficial effects on axonal growth reported by Hervera et al. were centered in sensory fibers after dorsal hemisection. Thus, potential differences in the outcome may rely on motor neurons, which have not been fully covered in the current studies.
- The HDAC sections are extremely detailed, while sections such as SIRT1 (3.2.3) and BRG1 (3.3.2) seem relatively short. Try to balance the depth.
R: We would like to clarify that SIRT1 is also a histone deacetylase, and the level of detail provided and extension is similar with that to other histone deacetylases in the review. Regarding BRG1, we agree that this section is comparatively shorter. However, the number of reports focused on this target in the context of nervous system injury and regeneration is currently very limited. As a result, it is not feasible to expand this section to the same depth. We have nonetheless ensured that the most relevant and up-to-date findings are included.
- The schematic diagrams are visually clean but lack panel labels or explanatory legends that could help readers decipher mechanisms between cell types.
R: Thank you for your suggestion. We have addressed this issue and modified figure 2 by adding more panel labels. We have also made a new Figure 3 with more details.
- The dual nature of BET inhibition (inflammation good, neurite growth bad) is presented, but no attempt is made to resolve or account for this paradox.
R: We have tried to address this issue in the text but, as suggested by the reviewer, we did not address it with sufficient depth. We have now included some sentences indicating that BET inhibition promotes axonal growth, in part through the expression of anti-inflammatory cytokines and STAT6 activation in glial cells. We have also indicated that BET inhibition reduces prolonged inflammation, which is detrimental for regeneration.
- The conclusion mentions briefly delivery and timing concerns, but the review would be strengthened by discussing in greater detail the translational failures of HDAC inhibitors and BET inhibitors in the clinic.
R: We would like to discuss further the translational failures of HDAC and BET inhibitors in the clinic, however these targets have been used in cancer therapies and not after PNI and SCI. Thus, mentioning the possible reason for failures in other diseases is not under the scope of this review. Additionally, as we mentioned in the conclusions, most of the studies in clinics have not made specific targeting of these epigenetic targets. Which may justify the observed side effects.
However, clinical trials are starting to include epigenetic studies. For instance, we found a reported trial ‘NCT06537427 - The Crosstalk Between the Epigenome and Mitochondria in SCI (CEM-SCI)’, although it is not yet recruiting patients. Thus, it will be possible to include clinical information soon.
- Table 3 uses abbreviations like CGNs and PNI without first defining them, define all acronyms upon first use in text and tables.
R: The acronyms in table 3 are detailed in the text of the heading.
‘Table 3. HATs involved in axonal regeneration. The table summarizes histone targets, effects on gene expression, as well as effects on regeneration, based on findings from in vivo and in vitro studies. CGN, cerebellum granular neurons; DRG, dorsal root ganglia; PNI, peripheral nerve in-jury; SCI, spinal cord injury.’
- Line 105: "as well as in distinct transcription factors" could be rephrased for clarity. Suggest proofreading in general.
R: We change the line to ‘and in transcription factors. It is important to clarify that the specific HATs and HDACs targeting of histones or other specific proteins are mentioned later in the text. Thus, it is not worth to enter in further detail at this point
- The review mainly cites animal studies. State briefly whether there are any relevant human postmortem or iPSC-derived neuron studies.
R: As the reviewer mentioned, most epigenetic studies are centered on animals or primary cultures. We did not find information about post-mortem studies. However, we did include articles in the section of astrocytes that work with human cells.
- Saha RN, Jana M, Pahan K. MAPK p38 regulates transcriptional activity of NF-kappaB in primary human astrocytes via acetylation of p65. J Immunol. 2007;179(10):7101-9.
- Berryer MH, Rizki G, Nathanson A, Klein JA, Trendafilova D, Susco SG, et al. High-content synaptic phenotyping in human cellular models reveals a role for BET proteins in synapse assembly. Elife. 2023;12.
- The section on astrocytes (3.4.3) is not well developed. Given the significance of glial scarring to CNS repair, the role of histone modifications in reactive astrocytes needs to be discussed more completely.
R: We agree with the reviewer, and have largely expanded the astrocyte subchapter, now subsection 4.3. Their role after SCI injury is crucial and actually deserve to be scrutinized in more detail.
- The final part may contain a schematic diagram or decision tree of possible therapeutic approaches that are directed by epigenetic targets and by neuronal subtypes.
R: To clarify the differences of distinct neuronal subtypes we have modified figure 2 and also included a third figure that is centered specifically in non-neuronal cells.
Comments on the Quality of English Language
Certain sentences are too lengthy or reiterate details provided previously. Condensing them would enhance readability.
R:We appreciate the reviewer’s comment and have carefully revised the manuscript to improve clarity and readability. Redundant or overly long sentences have been condensed to ensure a more concise and accessible presentation of the contents.

Round 2
Reviewer 2 Report
Comments and Suggestions for Authors
No further comments